# Assessing Electroencephalography as a Stress Indicator: A VR High-Altitude Scenario Monitored through EEG and ECG

**DOI:** 10.3390/s22155792

**Published:** 2022-08-03

**Authors:** Vasileios Aspiotis, Andreas Miltiadous, Konstantinos Kalafatakis, Katerina D. Tzimourta, Nikolaos Giannakeas, Markos G. Tsipouras, Dimitrios Peschos, Euripidis Glavas, Alexandros T. Tzallas

**Affiliations:** 1Human Computer Interaction Laboratory (HCILab), Department of Informatics and Telecommunications, University of Ioannina, Kostakioi, 47100 Arta, Greece; v.aspiotis@uoi.gr (V.A.); a.miltiadous@uoi.gr (A.M.); k.kalafatakis@qmul.ac.uk (K.K.); ktzimourta@uowm.gr (K.D.T.); giannakeas@uoi.gr (N.G.); eglavas@uoi.gr (E.G.); 2Faculty of Medicine, University of Ioannina, 45110 Ioannina, Greece; dpeschos@uoi.gr; 3Institute of Health Science Education, Barts and the London School of Medicine & Dentistry, Queen Mary University of London (Malta Campus), VCT 2520 Victoria, Malta; 4Department of Electrical and Computer Engineering, Faculty of Engineering, University of Western Macedonia, 50100 Kozani, Greece; mtsipouras@uowm.gr

**Keywords:** virtual reality, EEG, ECG, stress, high-altitude exposure, Occipital Alpha Asymmetry, HMD, Frontal Alpha Asymmetry, BPM, Perceived Stress Scale

## Abstract

Over the last decade, virtual reality (VR) has become an increasingly accessible commodity. Head-mounted display (HMD) immersive technologies allow researchers to simulate experimental scenarios that would be unfeasible or risky in real life. An example is extreme heights exposure simulations, which can be utilized in research on stress system mobilization. Until recently, electroencephalography (EEG)-related research was focused on mental stress prompted by social or mathematical challenges, with only a few studies employing HMD VR techniques to induce stress. In this study, we combine a state-of-the-art EEG wearable device and an electrocardiography (ECG) sensor with a VR headset to provoke stress in a high-altitude scenarios while monitoring EEG and ECG biomarkers in real time. A robust pipeline for signal clearing is implemented to preprocess the noise-infiltrated (due to movement) EEG data. Statistical and correlation analysis is employed to explore the relationship between these biomarkers with stress. The participant pool is divided into two groups based on their heart rate increase, where statistically important EEG biomarker differences emerged between them. Finally, the occipital-region band power changes and occipital asymmetry alterations were found to be associated with height-related stress and brain activation in beta and gamma bands, which correlates with the results of the self-reported Perceived Stress Scale questionnaire.

## 1. Introduction

Stress is a fundamental and ubiquitous part of human daily life. Individuals under stress may enter a hyperarousal state, combined with compensatory changes in their respiratory rate, muscle tone, or heart rate while their bodies undergo a series of subconscious neuroendocrine adaptations [1]. In the short term, the mobilization of the stress system may exert positive effects, such as increasing motivation or attention or enhancing goal-oriented behavior [2], but in the long-term, stress is also linked to negative effects that significantly impact physical and mental health, leading to memory dysfunction and mood disorders [3,4]. Scientific exploration of stress, amongst various methods, involves examining different biomarkers such as cortisol levels, heart rate indicators, galvanic skin response, pupil diameter, and other indicators [5] after a specific stressor has been induced. Designing such experimental processes can be demanding; the method of assessment and the choice of stressful cues is of crucial importance for producing reliable, quantifiable, and reproducible results. Studies have adopted multiple methods for inducing stress, including performance, psychological, or social tests such as the Trier Social Stress Test (TSST) [6] and the Maastricht Acute Stress Test (MAST) [7], each with different advantages and disadvantages [8].

Advancements in the field of Virtual Reality (VR) and Head-Mounted Displays (HMD) have been utilized by researchers investigating targeted stressors, for triggering specific stress indicators. HMDs offer feasibility, repeatability, and control while leaving space for the development of tangible innovative ideas. Highly realistic virtual environments that recreate real-life scenarios (e.g., train/roller coaster, escaping life-threatening events, reacting to emergencies, high-altitude exposure) or completely fantastic ones or replicate traditional stress tests (e.g., TSST, MAST) into the virtual dimension have been developed and validated [9,10].

As the human brain is a key component of the stress system [11], monitoring its activity via electroencephalography (EEG) can unravel details of acute stress responses and unveil new detection methods. EEG analysis methods have been extensively used in the study of neurodegenerative diseases [12], cognitive disorders [13] or Brain–Computer Interface (BCI) applications [14]. However, EEG measurements have not been comprehensively utilized in stress research, contrary to other neurophysiological data [15]. Stress-related studies using EEG commonly perform brain activity analysis or automatic classification. Brain activity analysis methods include functional and effective connectivity [16]. The former represents time-related coherence between neuron activities. Such studies explore and apply analysis techniques with interesting results; however, no specific guidelines exist regarding the choice of EEG features and their integration [4].

In that sense, the calculation of brain region asymmetry is often used as a functional connectivity measure. Various research works suggest that a functional lateralization in the frontal cortex is related to affective processing [17] and, in particular, may be an indicator for physiological stress [18]. Specifically, Frontal Alpha Asymmetry (FAA) is a measure used in a variety of protocols designed to evaluate mental stress. FAA is calculated from the difference of the logarithmic values of two symmetric frontal EEG electrodes (usually F3-F4 or F7-F8). Right-side frontal prevalence is related to negative emotion regulation and social withdrawal behaviors, while left-side prevalence is connected to positive emotions, superior emotional flexibility, and more effective emotion regulation [18,19].

Over the last five years, there have been significant advances in stress assessment through VR environments. Nonetheless, only a few have utilized multiple biomarkers including EEG to evaluate stress [20,21,22,23]. Stolz et al. utilized a VR room with avatars with different facial expressions and different sounds within a threat-conditioned context and used Event-Related Potentials (ERP) of EEG to investigate cortical processing [21]. In other research, Fadeev et al. performed a small-scale ad hoc study examining stress, utilizing multiple VR scenarios and observing the EEG, respiration rate, and heart rate alterations [22]. Wang et al. [23] used VR Richie’s Plank Experience and obtained EEG recordings. They then performed classification of the subjects based on their subjective evaluation of fear of heights by utilizing their EEG signal. However, to the best of our knowledge, there are only a handful of studies investigating—within an immersive VR scenario—the correlation between EEG and ECG-related bioindicators for stress such as BPM [24,25].

In this study, we aim to analyze brain neurodynamic correlations between the human stress mechanism and a quantifiable cardiovascular biomarker by exploiting VR HMDs to simulate an otherwise hazardous type of stressor. We explore whether the activation of different brain areas, when exposed to a VR high-altitude scenario, is related to stress, while using BPM as a validator. Additionally, we deploy the Perceived Stress Scale questionnaire to acquire the individuals’ subjective self-assessment of stress. In order to further investigate how high-altitude related stress may differently manifest among individuals, we created two groups of interest and herein compare the EEG biomarkers between them, resulting to useful insights. Such findings can unravel the brain’s integration in the stress mechanism and contribute to the development VR Exposure Therapy (VRET) methods, affective computing techniques, and diagnosis of stress-related disorders.

## 2. Materials and Methods

In this section, we describe the experimental protocol of this research, the EEG and electrocardiographic (ECG) data preprocessing, the feature extraction procedure, and the statistical methods used for the analysis of the participants. A complete flowchart of the experiment can be found in Figure 1. Our methodology includes three basic stages. The Recording, the Preprocessing, and the Analysis stage. The former is the experimental and signal acquisition process stage, which is explained in detail in Section 2.2. In the second stage, the signals are properly handled to be analyzed later by implementing techniques such as artifact removal, heartbeat per minute extraction, etc. The details of this stage are laid out in Section 2.3. The latter stage incorporates signal analysis techniques and statistical methodologies that investigate our scope/hypothesis and lead to our results and conclusions. Those can be found in Section 2.4 and Section 2.5.

### 2.1. Subjects

Twenty-one participants (ranging from 20 to 27 years of age; 8 females and 13 males) with normal or corrected to normal vision were chosen for this experiment. None of the participants were familiar with the VR scenario (Richie’s Plank Experience steam game, reference) that was used, and they had minimal to no previous experience with HMDs. Participants were informed that the experiment was related to a VR experience. However, the nature of the experiment and the stress assessment research goal were not disclosed to them. Only 18 of the 21 recordings were deemed appropriate to be used in the study.

### 2.2. Experimental Protocol and Data Acquisition

A Meta Quest 2 VR device was used for the VR stimulus, and a DSI-24 wearable EEG device with 21 electrodes captured the EEG recordings during the experiment. Quest 2 VR headset offers a 1920 × 3664 resolution with 773 PPI and a frame rate of 60–120 Hz. The safety of use was ensured by the built-in cameras in the front side that do not allow the participant to walk outside the designated area. DSI-24 is a wireless EEG headset with dry electrodes manufactured by Wearable Sensing, San Diego, CA, USA. The electrodes Fz, F3, F4, Cz, C3, C4, T7, T8, Pz, P3, P4, P7, P8, O1, O2, A1, A2 were placed according to the 10–20 international standard. However, four electrodes that were originally designated for the positions F7, F8, Fp1, Fp2 were relocated upwards so that the VR and EEG headset could properly fit. These electrodes were excluded from this study. Two electrodes placed on the sternum under the heart were used for recording the ECG activity. The ECG electrodes were part of the DSI-24 bundle and were connected directly to the headset. Therefore, there was no need for synchronization of the EEG and ECG signals. The sampling rate was 300 Hz and all electrode impedances were below 5 KΩ for the whole duration of the study. The EEG signals were recorded with the Cz electrode serving as ground.

Richie’s Plank Experience steam game is a VR game that makes the participant enter an elevator, takes them to the top floor of a skyscraper, and allows them to walk over and finally jump off a plank extending over the edge of the building. In this experiment, a wooden plank with same dimensions as the virtual one was fixed on the floor so that the participants could walk on it. First, the devices (EEG, ECG and HMD) were placed on the participants. They were given time to familiarize themselves with the equipment and later were asked to close their eyes while standing up for 1–2 min, during which a resting state recording was obtained. Then, they were asked to open their eyes, where they would find themselves in the virtual environment, inside an elevator with the door open, on the ground floor of a building. They were urged to look around them and explore their surroundings without leaving the elevator, with the EEG and ECG devices recording for whole duration of the experiment. Next, they were instructed to press a button inside the elevator. The door closed, the elevator moved to the top floor, and the door opened again. At that point, the participants were told that they should walk on the plank in front of them (at that time, they would realize the actual plank laying in front of them). When they reached the far side of the plank, they were asked to jump. This part of the experiment—after the elevator door opening, walking on the plank, and finally being asked to jump off the building—was considered the stressful stimuli. At the end of the experimental part, the participants were informed that this experiment was about assessing stress system mobilization through VR HMD and were asked to fill the Perceived Stress Scale questionnaire [26]. Table A1 in the Appendix section contains the results of the PSS questionnaire for every participant along with their age, gender, and previous VR experience. Figure 2 illustrates the experimental design of this study.

#### Perceived Stress Scale Questionnaire

The Perceived Stress Scale (PSS) [26] is a well-established psychological tool for quantifying the perception of stress. It is equipped with quality psychometric properties that can measure the self-reported levels of stress experience. It includes questions about the severity and frequency of stress-related thoughts and feelings aiming to quantify the level of subjective stress perception of an individual. In this study, PSS-10 was used, in which the answers to 10 queries produce an overall score that ranges from 0 to 40, with low scores indicating lower stress and high scores indicating higher perceived stress. Scores from 0 to 13 can be considered as low stress, 14 to 26 as moderate stress, and 27 to 40 as high stress.

### 2.3. Data Preprocessing

The EEGLAB Matlab Toolbox was used for the preprocessing stage [27]. EEG recordings were re-referenced to the A1, A2 electrodes which were placed on the mastoids. A fourth-order Butterworth band-pass filter was applied allowing frequencies between 0.4 and 48 Hz. EEG signals were split to three different files: resting state, calm state, and stressed state. Artifact rejection was performed by using the Artifact Subspace Reconstruction (ASR) and the Independent Component Analysis (ICA) method (FastICA algorithm [28]). For the ASR, a conservative threshold of 17 was chosen as the maximum acceptable 0.5-second-window standard deviation. For the ICA, components that were classified as eye or muscle artifacts with a possibility of 0.9 or above were automatically rejected. Figure 3 presents four different Independent Components as classified by the automatic classification routine “ICLabel” in the EEGLAB platform. The first two components are classified as eye artifacts and the third component is classified as muscle artifact. These components were removed. The fourth scalp heatmap represents a brain activity component that is not removed.

Subsequently, the signal was epoched in 4-second windows, and the Power Spectral Density (PSD) of each frequency band at each electrode was calculated using the Welch [29] method. The frequency bands were defined as follows:Delta: 0.5–4 HzTheta: 4–8 HzAlpha 8–13 HzBeta: 13–25 HzGamma: 25–45 Hz

Finally, each frequency band was averaged across the electrodes for each cortex of the brain, leading to the calculation of the average absolute power for each brain region.

The ECG signal preprocessing consisted of the following steps. First, a FIR filter was applied. Next, a peak enhancement function was used to normalize the amplitude and increase the R-peak amplitude in comparison with the rest of the signal. To perform the R-peak detection, an adaptive peak detection threshold was set. After the peaks were detected, a sliding window of 6 seconds was applied to calculate the BPM for each time point. Finally, 3-second-window averaging was applied to the BPM signal for smoothing. The ECG signal was also segmented into three parts: resting state, calm state, and stressed state.

### 2.4. Feature Extraction

In this section, the EEG and ECG metrics calculated across each subject are presented.

#### 2.4.1. Brain Region Power

Regarding the EEG, the power of each band that was calculated for each 4-second window was averaged for the three distinct states. Then, the average band power for each brain region was calculated. The brain regions are defined as follows:Occipital = {O1,O2}Temporal = {T3, T5, T6, T4}Parietal = {C3, Cz, C4, P3, Pz, P4}Frontal = { F7, F3, F4, F8}

The difference of band power across each region and each band was calculated as
(1)$${\mathrm{Band}}_{\mathrm{Region}}={\mathrm{Band}}_{\mathrm{Region}\;\mathrm{at}\;\mathrm{stressed}\;\mathrm{state}}-{\mathrm{Band}}_{\mathrm{Region}\;\mathrm{at}\;\mathrm{calm}\;\mathrm{state}}$$
where the Power Spectral Density (PSD) of each band is computed for each discrete 4-second window and then averaged for each state. To estimate the PSD with the Welch method, the signal was segmented in non-rectangular windows using the Hamming method. Thus, for L time windows, the periodogram of each window is defined as [30]
(2)$$Y_{i}\left(\omega \right)=\frac{1}{\mathrm{PQ}}{\left|\overset{P-1}{\underset{n=0}{\sum }}x_{i}\left(n\right)c\left(n\right)e^{-j\omega n}\right|}^{2}$$

The average energy of each window Q is
(3)$$Q=\frac{1}{P}\overset{P-1}{\underset{n=0}{\sum }}c^{2}\left(n\right)$$

#### 2.4.2. Asymmetry Measures

Frontal Alpha Asymmetry (FAA) and Occipital Alpha Asymmetry (OAA) scores were calculated. FAA is a metric widely used to express the asymmetry of the frontal cortex, expressed as
(4)FAA=log(F4)−log(F3)OrFAA=log(F8)−log(F7)

A positive value of FAA indicates more right-sided alpha power. Research suggests that right-sided alpha power denotes more left hemispheric activation and vice versa [31]. Therefore, a positive FAA value indicates left hemisphere activation. In this study, the F4–F3 combination was implemented for measuring the asymmetry score. For each time window, after the PSD was calculated, the FAA score was measured. The FAA scores of every time window for each state were averaged to create a total FAA score for each condition. 

A similar process was performed for the calculation of the Occipital Alpha Asymmetry (OAA), defined as
(5)OAA= log(O2)−log(O1)

#### 2.4.3. Heart Rate Measures

The average BPM for each stage of the study (resting state, calm state, stressed state) was calculated across all subjects after the BPM signal was extracted from the ECG signal, following the methodology explained in Section 2.3. BPM serves the purpose of validating the existence of stress due to the fact that there are no other factors that affect the participants during the experiment apart from virtual high-altitude exposure. There is no increase in physical activity nor any other environmental changes such as temperature between the different stages of the experiment. In addition, an increase in heart rate has been confirmed to be a reliable indicator of mental stress by multiple research works [32,33,34,35].

### 2.5. Statistical Analysis

A series of statistical tests was performed to evaluate the results of the experiment. In order to examine whether the changes of each biomarker between the stressed and calm states were significant, a paired *t*-test analysis was conducted after examining the distribution normality with a Kolmogorov–Smirnov test. Next, the participants were split into two groups based on their BPM alteration and in-group paired *t*-tests were performed to examine the significance of the Asymmetry Score alterations. Moreover, the non-parametric Mann–Whitney U-Test was performed to investigate if the difference of the PSS results between the two groups was significant. Furthermore, the correlation matrix for the BPM increase (from calm to stressed), the brain region energies increase, and the frontal and occipital asymmetry alterations was produced via Spearman correlation analysis. Spearman correlation analysis was also employed for exploring the connection between the PSS and the EEG and ECG biomarkers.

## 3. Results

In this section, we illustrate the dissimilarity between the different states and present the results of the statistical analysis. Figure 4 represents an indicative BPM diagram of a participant throughout the experiment. The blue and orange line is the BPM signal before and after smoothing, respectively. Each colored area represents the time duration of the three distinct states (resting, calm, stressed). It can be observed that the participant had increased heart activity at the stressful state (the light blue area), demonstrating that being at the top floor of the skyscraper can indeed be a stressful experience.

Herein, the alpha, beta, and gamma band powers are compared across the calm and stressed states for all brain regions. Figure 5 illustrates the BPM comparison across all subjects between the calm state (ground floor of the elevator) and the stressed state (top floor). It also represents the comparison of the alpha, beta, and gamma bands across each brain region. Our data show that the occipital area was engaged significantly more than other brain regions during the stressful condition.

In order to examine whether the alteration of the ECG and EEG biomarkers was significant, statistical analysis was performed. The biomarkers evaluated were alpha, beta, and gamma power for each of the frontal, parietal, temporal, and occipital brain regions and OAA, FAA, and BPM. Specifically, a Kolmogorov–Smirnov test was employed and validated that each of the biomarker values was normally distributed. However, the Kolmogorov–Smirnov test did not validate the normal distribution of the PSS results. Next, we applied a paired *t*-test to verify whether the alteration of each marker between the calm and stressed states was statistically significant. The results of the paired *t*-test are presented in Table 1. The change in alpha, beta, and gamma band of parietal, temporal, and occipital, respectively, as well as the BPM were found to be statistically significant since the two-sided *p* value of the paired *t*-test was <0.05. Alpha, beta, and gamma power band changes were not statistically significant in the frontal region; however, frontal alpha and frontal beta showed one-sided *p* values < 0.05. The BPM change was found to be statistically significant, as expected. However, FAA and OAA changes did not produce any statistical significance.

### 3.1. Group Analysis

Figure 5 indicates the need for examining whether this activation was related to stress caused by visual stimuli. To do this, we created two groups of participants. The first group consisted of subjects with a normal baseline heart rate (BPM < 100) which was not increased by more than 13 BPM during the stressful condition. The second group consisted of subjects with a normal baseline heart rate which increased by more than 13 BPM during the stressful condition; the threshold of 13 BPM (0.22 Hz) was chosen after studying the available literature [36]. A BPM increase of more than 13 was considered a significant increase, while an increase smaller than 13 was considered non-significant. Group 1 consisted of seven participants. Group 2 consisted of nine participants. Two participants were left out because of high baseline BPM. Figure 6 is a scalp heatmap comparison of the brain activity between a participant from Group 1 with one from Group 2. Different colors represent the difference from the average Power Spectral Density of the brain for each band expressed in 10 × log_10_ (uV^2^/Hz). The heatmap limits are from −8 (deep blue) to +8 (deep red). Figure 7a represents the comparison of the BPM increase between the two groups. Figure 7b represents the comparison of the occipital activity of the two groups. This comparison points out that occipital activity may be an indicator for stress system activation due to visual stimulation.

The change in OAA and FAA scores was not statistically significant, as illustrated in Table 1, when a paired *t*-test was performed for the entirety of the participant pool. Nonetheless, a paired *t*-test was re-employed for the two groups separately. The results are presented in Table 2. Group 1 (participants with no BPM increase) did not present any statistically important changes in OAA or FAA between the calm and stressed states. In contrast, Group 2 (participants with BPM increase > 13) presented statistically significant changes in OAA but not in FAA. Figure 7c illustrates how the asymmetry scores changed between the two states for the two groups. Group 2 had generally more left alpha activation during the stressed state compared to Group 1, where the asymmetry difference was near 0 (left Alpha Asymmetry suggests right-side occipital activation).

The non-parametric Mann–Whitney U-Test for PSS results was performed since the Kolmogorov–Smirnov hypothesis of normal distribution for the PSS scores could not be accepted. The results are presented in Table 3. No statistically important differences were observed between the two groups.

### 3.2. Correlation Analysis

The variation of each power band has been examined for whether it is correlated with the variation of BPM by using the Spearman Correlation method. Table 4 represents these correlations. The change in the absolute power of alpha, delta, and theta band in the occipital region is strongly correlated with the change of BPM (Spearman Correlation value ≥ 0.5). Alpha power change in the temporal region was also significantly correlated to the BPM, with the Spearman Correlation value reaching 0.47.

A Spearman Correlation analysis was also employed for the PSS scores. In particular, the correlation between PSS score and the alteration of the biomarkers (namely BPM, frontal, temporal, parietal, and occipital power in delta, theta, alpha, beta, and gamma bands, respectively) between the calm and stressed states was calculated. The results can be found in Table 5. Frontal beta power increase appeared to be strongly correlated to PSS score (0.5 Spearman Correlation score). Parietal beta and gamma power increase was significantly correlated to PSS score (0.56 and 0.71). Temporal beta and gamma power increase was significantly correlated to PSS score (0.7 and 0.6). Occipital beta and gamma power increase was also significantly correlated to PSS score (0.53 and 0.56). However, the BPM alteration was not found to be correlated to the PSS score (0.058).

## 4. Discussion

This study investigated the EEG biomarkers of the human brain under stressful conditions deriving from a high-altitude exposure VR scenario. Heart rate measurement was employed as a validator of the existence of stress. The band power of each brain region and the activation asymmetries of the brain were explored. Furthermore, the results of the Perceived Stress Scale questionnaire were assessed.

Over the last decade, multiple studies have used VR environments to evoke emotional reactions [22,37]. Eye tracking [38], questionnaires [39], and respiration signals and ECG [40] are the most common measures used to assess stress in VR-related studies, while EEG has seldom been used [41]. One key explanation for this is that the combination of an EEG and a VR headset is not easily achievable due to device placement overlapping restrictions. Hu et al. performed a study that also used the software Richie’s Plank Experience to classify the participants’ fear level through EEG [41]. Fadeev et al. performed an analysis of different VR scenarios on three subjects with abnormally intense emotional responses due to health conditions [22]. However, to the best of our knowledge, no research had (both of) these two key elements from our study: (1) a well-established in literature biomarker to confirm the existence of stress, (2) EEG recording throughout the whole duration of the experiment (rather than recording at standard checkpoints). 

Moreover, there are multiple studies that used EEG signals for mental stress assessment, as reported by Katmah et al. [4]. Connectivity methods and spectral and asymmetry characteristics have been studied as stress markers. Frontal Alpha Asymmetry and frontal power have also been used as stress indicators [5,42]. However, studies that imported FAA or frontal activity to evaluate stress used mathematical or social ordeals as the stressful experience [43]. In this study, stress is related to visual stimuli, so higher occipital activity was expected. Thus, we employed an uncommon asymmetry measure, Occipital Alpha Asymmetry, and observed whether it is connected to stress.

Frontal Alpha Asymmetry (representing relative stronger neural activity of the left frontal cortex over its right counterpart) is considered in psychological research as a concurrent and prospective marker of affective processing, most commonly treated as either a predictor or an outcome variable related to motivation, emotion regulation, and psychopathology [44]. Nevertheless, this marker is most likely linked to complex neural dynamics that involve large-scale brain networks, as well as complex psychological mechanisms, and thus, its replication in different experimental settings/study populations will not be consistent. Any relationship between FAA and the stress system mobilization is under investigation. Current pieces of evidence do not link FAA with other markers of stress induction [17,19], while resting FAA has failed to be considered as a reliable marker of post-traumatic stress disorder [45]. Similarly, in our study, no presence of FAA was evident during the stressful part of the experiment.

On the contrary, in this study, we observed the presence of Occipital Alpha Asymmetry during the stressful part of the experimental process. The meaning of this observation is unclear and prompts further investigation, since only a few sources of evidence exist on the physiological or psychological significance of OAA. Nevertheless, OAA has been linked to the mobilization of the behavior inhibition system (BIS), i.e., a neuropsychological system that predicts an individual’s response to anxiety-relevant cues in a given environment [46]. In this context, we could hypothesize that BIS was recruited during the stressful part of the experimental process (especially the parts where subjects needed to walk on the narrow plank and were subsequently urged to jump from such a high altitude).

Recent meta-analytic data support the notion that heart rate variability (HRV) in the context of stress system mobilization may indicate the degree to which a higher-level cortical “core integration” system is integrated with the brainstem nuclei that directly regulate the heart [47]. In other words, stress-related differences in the HRV may reflect differences in the neural processing-associated responses to stressful insults. We thus chose to divide our study sample on the basis of different stress-induced heart rate responses and explore whether the EEG markers differed between the two groups. The data (showing strong correlations between the occipital EEG rhythms and the heart rate, as well as differences in the occipital EEG activity and the OAA between the two groups) further support the idea that these changes in EEG-recorded neural activity of the occipital regions are related to stress and are also potentially linked to trait-like features.

The correlation analysis between the PSS scores and the EEG biomarkers revealed some interesting results. There seemed to be a strong connection between the subjective self-reported stress quantification of the participants and the increase in the high-frequency brain activity in all brain regions, meaning that the participants that perceived themselves as more easily stressed individuals indeed showed increased brain activity during the stressful part of the experiment. Such results could further support the validity of using EEG biomarkers as stress detectors and should be taken into consideration in future research, especially when considering that there is more evidence in previous research that PSS score and EEG band power under stress can be correlated [48]. However, these findings cannot stand as credible on their own, for the following reasons. Firstly, we did not observe a significant difference between the PSS score of the two groups, and at the same time, there was no correlation between the BPM and the PSS score (BPM being the established stress validator in this experiment). Secondly, the size of the participant pool of this experiment may have impacted the validity of the distribution of the self-ratings (the same experiment on a different group of people may or may not produce the same results).

At this point, we should address other limitations of the present work. The interconnectivity of the EEG device and the VR headset was not intuitive. In order to ensure that both devices were placed properly on all participants, we had to exclude the four frontal electrodes. Also, due to the dimension specification of the devices, participants with a head circumference less than 54 cm could not be recruited. In addition, due to the time restrictions of this experiment, the participant pool was limited. Furthermore, other markers related to stress, such as HRV features, were not calculated in this stage of our research.

More research on EEG as a stress indicator is included in our future research plans. Specifically, it is within our intentions to incorporate HRV features in our methodology in order to unveil any possible correlations between EEG features and stress mechanisms. Furthermore, a methodology for automatic real-time detection of stress from EEG signals using novel Machine Learning techniques will be proposed. By expanding the participant pool, we could also use the Perceived Stress Scale questionnaire scores to propose a regression scheme for accurately predicting the Stress Scale of an individual. Finally, we aspire to design or include more Virtual Environment stressors in our approach to study other stress manifestations.

## 5. Conclusions

In this study, we attempted to assess the impact of stress on EEG when using a VR high-altitude exposure scenario. EEG and ECG signals were obtained from all participants throughout the experiment. BPM was used as a validator of the appearance of stress on participants, while various EEG features such as brain region absolute power and Frontal and Occipital Asymmetry scores were evaluated as stress indicators. The notion of using VR experiences as stressors is supported by our results, and useful outcomes regarding EEG functional operation have emerged. FAA activation under stress could not be validated by our study; however, we observed increased activity in the occipital brain region as well as OAA in the group considered to be stressed. Additionally, the observation that the increase in energy of the delta, theta, and alpha bands of the occipital region was strongly correlated with the increase in BPM points to the need for further research on the connection between the type of stressor and brain activation. Finally, a strong correlation was observed between PSS scores and the brain response for high-altitude exposure at beta and gamma bands.

## Figures and Tables

**Figure 1 sensors-22-05792-f001:**
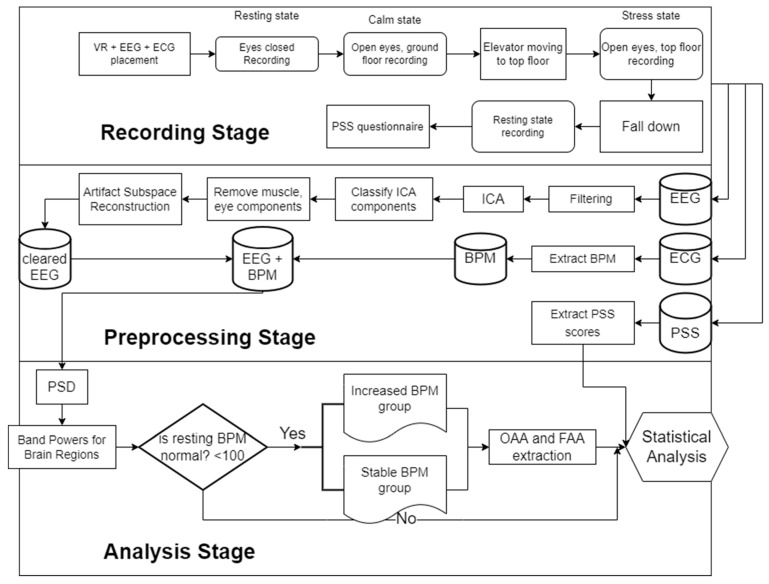
Flowchart of the experiment.

**Figure 2 sensors-22-05792-f002:**
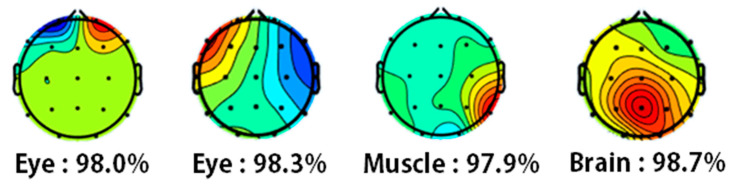
Independent components as classified by ICLabel.

**Figure 3 sensors-22-05792-f003:**
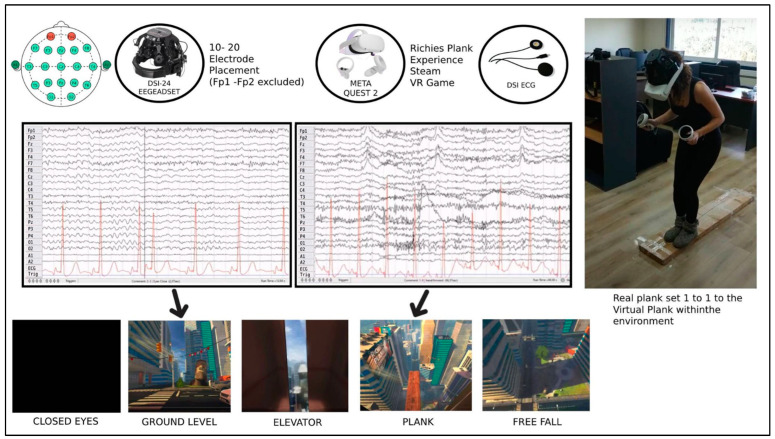
Experimental design.

**Figure 4 sensors-22-05792-f004:**
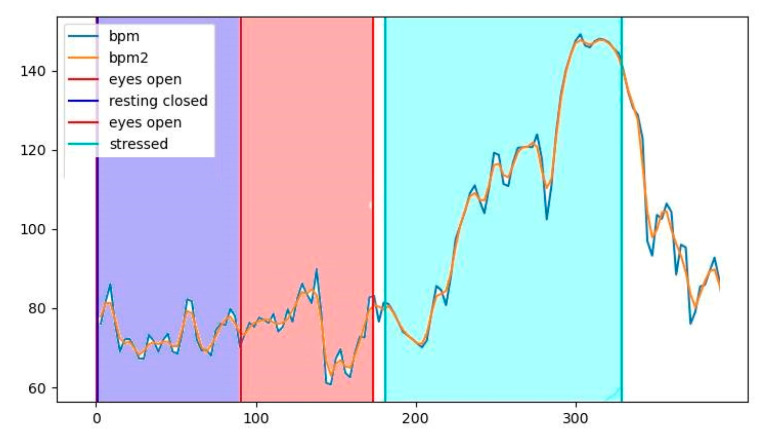
BPM signal of a subject across the experiment. Magenta represents the resting state, red represents the calm state, and light blue represents the stressed state.

**Figure 5 sensors-22-05792-f005:**
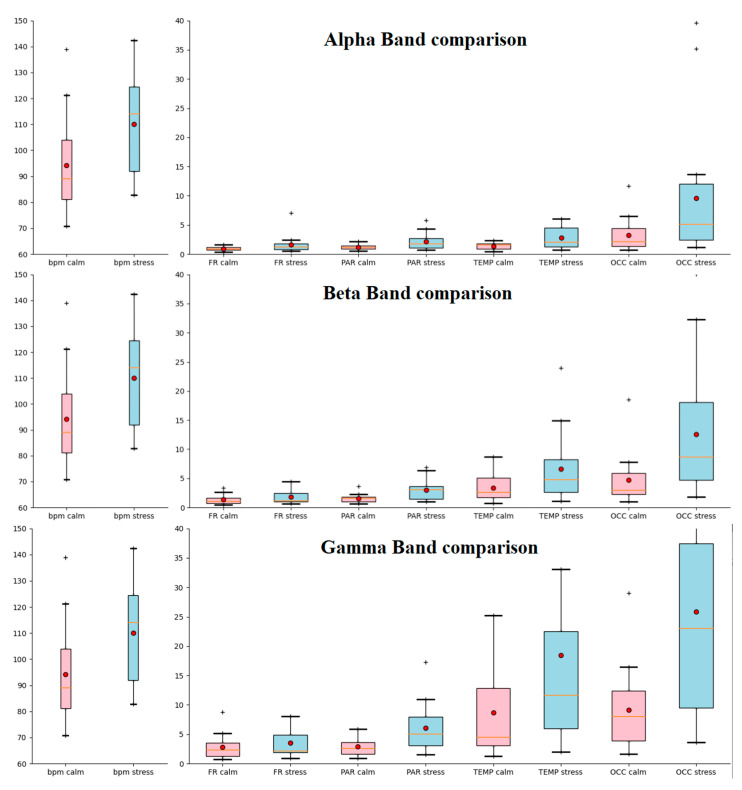
Calm and stressed state comparison of BPM and alpha, beta, and gamma bands. FR represents frontal cortex, PAR represents parietal cortex, TEMP represents temporal cortex, and OCC represents occipital cortex. The “+” sign at the top of the boxplots represents extreme values.

**Figure 6 sensors-22-05792-f006:**
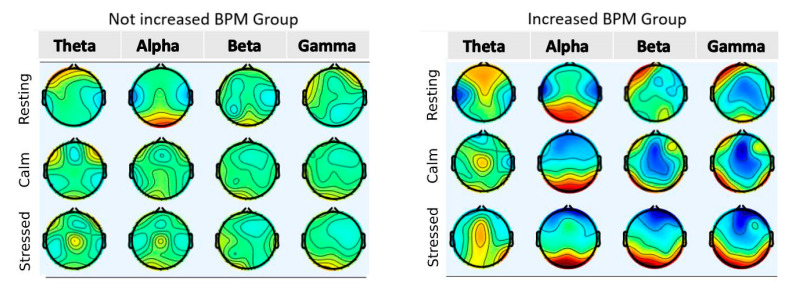
Brain heatmap comparison of subjects between the two groups for theta, alpha, beta, and gamma energy bands.

**Figure 7 sensors-22-05792-f007:**
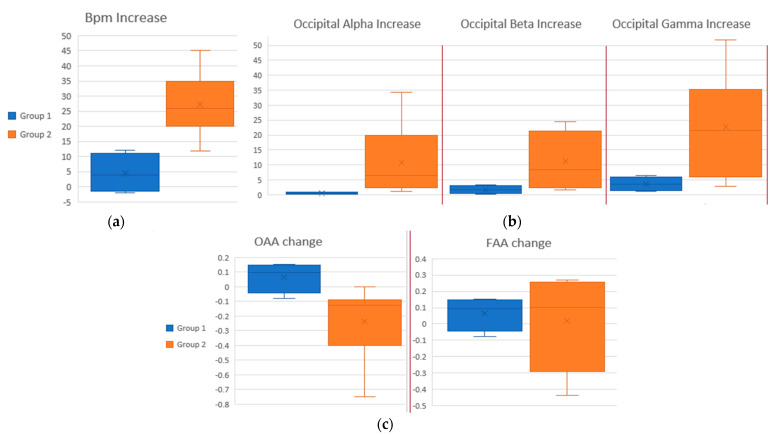
(**a**) Increase in BPM between groups. *Y*-axis is average BPM in stressed state minus average BPM in calm state. (**b**) Increase in occipital activity compared between groups. *Y*-axis is Power Spectral Density (PSD) calculated in uV^2^/Hz * 10^11^ in stressed state minus PSD in calm state. (**c**) OAA and FAA change compared between groups. *Y*-axis is OAA in stressed state minus OAA in calm state and FAA in stressed state minus FAA in calm state, respectively, calculated as mentioned in Section 2.4.2.

**Table 1 sensors-22-05792-t001:** Paired *t*-tests for each measure between the calm and stressed states. The * symbol indicates statistical significance with *p* value < 0.05.

	Calm-Stress	*t*	One-Sided *p*	Two-Sided *p*
Frontal	Alpha	−2.036	0.030 *	0.060
	Beta	−1.890	0.039 *	0.078
	Gamma	−1.239	0.117	0.234
Parietal	Alpha	−3.620	0.001 *	0.003 *
	Beta	−4.265	0.000 *	0.001 *
	Gamma	−4.359	0.000 *	0.001 *
Temporal	Alpha	−3.807	0.001 *	0.002 *
	Beta	−3.315	0.002 *	0.005 *
	Gamma	−3.039	0.004 *	0.008 *
Occipital	Alpha	−2.701	0.008 *	0.016 *
	Beta	−3.823	0.001 *	0.002 *
	Gamma	−4.506	0.000 *	0.000 *
	BPM	−4.327	0.000 *	0.001 *
	FAA	−0.599	0.279	0.557
	OAA	1.008	0.164	0.328

**Table 2 sensors-22-05792-t002:** Paired *t*-test in asymmetry scores for the two groups. The symbol * indicates statistical significance, with *p* value < 0.05.

	OAA Calm–Stressed			FAA Calm–Stressed		
	*t*	One-Sided *p*	Two-Sided *p*	*t*	One-Sided *p*	Two-Sided *p*
Group 2	2.733	0.015 *	0.029 *	−0.203	0.422	0.845
Group 1	−1.971	0.072	0.143	−1.269	0.147	0.294

**Table 3 sensors-22-05792-t003:** Independent samples *t*-test and Mann–Whitney U test for the PSS scores of the participants of the two groups. The differences did not prove to be statistically important in either test.

	PSS Score		*t*-Test Significance		Mann–Whitney U-Test						
	Mean	Std. Dev	One-Sided *p*	Two-Sided *p*	Sum of Ranks	Expected Sum of Ranks	Mean of Ranks	Expected Mean of Ranks	U-Value	Expected U-Value	Critical U-Value at *p* < 0.05
Group 1	13.42	4.03	0.208	0.416	84	76.5	9.33	8.5	24	31.5	12
Group 2	15.55	6.08			52	59.5	7.43	8.5	39	31.5	

**Table 4 sensors-22-05792-t004:** Spearman Correlation of the alteration of each power band with the alteration of BPM. The symbol * indicates high correlation (≥0.5).

Frontal				
Delta	Theta	Alpha	Beta	Gamma
0.16	0.18	0.37	0.23	0.24
**Temporal**				
Delta	Theta	Alpha	Beta	Gamma
0	0.26	0.47	0.32	0.31
**Parietal**				
Delta	Theta	Alpha	Beta	Gamma
0.17	0.079	0.44	0.31	0.22
**Occipital**				
Delta	Theta	Alpha	Beta	Gamma
0.64 *	0.5 *	0.55 *	0.44	0.43

**Table 5 sensors-22-05792-t005:** Spearman Correlation of the alteration of each power band and the alteration of the BPM with the PSS score. The symbol * indicates high correlation (≥0.5). The symbol ** indicates correlation ≥0.7.

Frontal				
Delta	Theta	Alpha	Beta	Gamma
0.22	0.41	0.39	0.5 *	0.52 *
**Parietal**				
Delta	Theta	Alpha	Beta	Gamma
0.31	0.22	0.38	0.56 *	0.71 **
**Temporal**				
Delta	Theta	Alpha	Beta	Gamma
0.35	0.48	0.38	0.7 **	0.6 *
**Occipital**				
Delta	Theta	Alpha	Beta	Gamma
0.11	−0.15	0.32	0.53 *	0.56 *
		**BPM**		
		0.058		

## Data Availability

Available upon request.

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
