# Peer review of "Assessing Electroencephalography as a Stress Indicator: A VR High-Altitude Scenario Monitored through EEG and ECG"

_sensors, 2022, doi:10.3390/s22155792_

Round 1

Reviewer 1 Report

The authors stated that their goal was to analyze the brain-neurodynamic correlates of human stress mechanisms and quantifiable cardiovascular biomarkers. While we understand the importance of this study, there are several flaws in the current study.

1. The stressors chosen by the authors should be experimentally confirmed to be highly stressful to the participants. An increase in the power ratio of a particular frequency band of the EEG does not necessarily mean that stress was induced. It is also helpful to introduce subjective assessments. The authors recorded the Perceived Stress Scale questionnaire. Why did they not use it in their analysis?

2. Please provide the rationale for choosing BPM as the ECG measure; several measures of HRV have been proposed (pNN50, rMSSD, SDNN, etc.), but only using BPM is less informative.

3. The author states that no studies have examined the relationship between EEG and ECG in VR environments, but it should be toned down.

[1] https://ieeexplore.ieee.org/abstract/document/9176022                                   [2] https://www.techrxiv.org/articles/preprint/Classification_of_VR-Gaming_Difficulty_Induced_Stress_Levels_using_Physiological_EEG_ECG_Signals_and_Machine_Learning/16873471

[3]https://www.mdpi.com/1424-8220/19/9/1991

4. The introduction section is unnecessarily too long; please consider shortening it.

Author Response

Reviewer 1

Comment 1: The stressors chosen by the authors should be experimentally confirmed to be highly stressful to the participants. An increase in the power ratio of a particular frequency band of the EEG does not necessarily mean that stress was induced. It is also helpful to introduce subjective assessments. The authors recorded the Perceived Stress Scale questionnaire. Why did they not use it in their analysis?

Response: We thank the reviewer for their valuable feedback. We leveraged the increase of the Heart Rate which is a widely used indicator of stress [1], [2] and monitored the EEG response (the increase of the EEG activity is not meant to be used as a validator of the existence of stress). Nevertheless, your insightful comment on the lack of clarification was addressed by further explaining the reasoning of this study at the paragraph 2.4 of Materials and Methods. Specifically, we created the sub-section 2.4.3 that describes the Heart Rate Measures and added the lines: Line 122-127 “The average BPM for each part of the study (resting state stage, calm stage, stressed stage) was calculated across all subjects, after the BPM signal was extracted from the ECG signal by the methodology explained in section 2.3. The BPM services the purpose of vali-dating the existence of stressor. Specifically, the increase of the heart rate can confirm the existence of stress, as there are no other factors that affect the participants during the experiment, apart from virtual high altitude exposure. There is no increase in physical activity nor other environmental changes such as temperature, between the different stages of the experiment. The increase in heart rate has been confirmed to be a reliable indicator of mental stress by multiple researches [34–37].”

We consider that the subjective measurements of the questionnaire is better used when the participant pool is bigger. Thus, we deemed the pool of 20 participants to be on the small side for a subjective measure to be reliable. We intend to use the results of the Perceived Stress Scale scoring in a following research that we will have acquired ~50 recordings from different participants. This research will use both the HRV measures and the PSS questionnaire results.

The Perceived Stress Scale is a measure that can often bring inconclusive results when the participant pool is small, considering that it is subjective. We intend to use the results of the Perceived Stress Scale scoring in a following research with more participants. However we mention it in this paper for clarity and reproducibility of the experimental process.

Comment 2: Please provide the rationale for choosing BPM as the ECG measure; several measures of HRV have been proposed (pNN50, rMSSD, SDNN, etc.), but only using BPM is less informative.

Response: We agree with the reviewer and we are aware that HRV measures have been also extensively used in the literature for stress monitoring. Nonetheless, the main focus of this research is the investigation of EEG related analysis in respect to stress. Since BPM is a sufficient and well-established biomarker of stress [1], [2] and it was not within our present scope to in-depth analyze the heart related biomarkers, we deemed not necessary to use more than one validators. Considering that this research is a part of a bigger research project related to VR and biomarkers, HRV will be employed in the future work.

Comment 3: The author states that no studies have examined the relationship between EEG and ECG in VR environments, but it should be toned down.

Response: We appreciate the insightful feedback and the references. Indeed, the suggested references [5], [6] do propose a methodology that employs EEG and ECG in a VR scenario (however they do not investigate the correlation between them). So, the statement has been toned down as asked by the reviewer. Also, the suggested references have been included.

Comment 4: The introduction section is unnecessarily too long; please consider shortening it.

Response: The introduction has been significantly reduced, according to the reviewer’s suggestion. The reviewed document is submitted in “track changes” .docx format, so every change can be seen in detail.

References

[1]         J. Taelman, S. Vandeput, A. Spaepen, and S. van Huffel, “Influence of Mental Stress on Heart Rate and Heart Rate Variability,” 2009, pp. 1366–1369. doi: 10.1007/978-3-540-89208-3_324.

[2]         J. M. Torpy, A. E. Burke, and R. M. Glass, “Acute Emotional Stress and the Heart,” JAMA, vol. 298, no. 3, p. 360, Jul. 2007, doi: 10.1001/jama.286.3.374.

[3]         W. E. J. Knight and N. S. Rickard, “Relaxing Music Prevents Stress-Induced Increases in Subjective Anxiety, Systolic Blood Pressure, and Heart Rate in Healthy Males and Females,” Journal of Music Therapy, vol. 38, no. 4, pp. 254–272, Dec. 2001, doi: 10.1093/jmt/38.4.254.

[4]         H. Steiner, E. Ryst, J. Berkowitz, M. A. Gschwendt, and C. Koopman, “Boys’ and girls’ responses to stress: affect and heart rate during a speech task,” Journal of Adolescent Health, vol. 30, no. 4, pp. 14–21, Apr. 2002, doi: 10.1016/S1054-139X(01)00387-1.

[5]         S. Pratiher et al., “Classification of VR-Gaming Difficulty Induced Stress Levels using Physiological ( EEG & ECG ) Signals and Machine Learning,” UMBC Student …, 2021, doi: 10.36227/techrxiv.16873471.v1.

[6]         M. Athif et al., “Using Biosignals for Objective Measurement of Presence in Virtual Reality Environments,” in 2020 42nd Annual International Conference of the IEEE Engineering in Medicine & Biology Society (EMBC), Jul. 2020, pp. 3035–3039. doi: 10.1109/EMBC44109.2020.9176022.

Reviewer 2 Report

The authors of the manuscript ‘Assessing Electroencephalography as a Stress Indicator: A VR High Altitude Scenario Monitored through EEG and ECG’ monitored 24 EEG and ECG biomarkers under high-altitude scenario stress.

 There are several questions as below about this manuscript:

 1. The detail information of the devices used in the study should be provided.

2. The author should provide the method of analysis.

3. In the Table 1, it is better that the authors label the significant result by using star sign instead of blackbody.

4. In Figure 5, what does the plus sign mean?

5. In Figure 7, the authors should provide better quality figures.

6. The degree of stress should be provided.

Author Response

Reviewer 2

Comment 1: The detail information of the devices used in the study should be provided.

Response: We thank the reviewer for the valuable comments. Detailed information about the DSI device and the Meta Quest 2 device have been included in the revised manuscript, as the reviewer asked.

“Quest 2 VR headset offers a 1920x3664 resolution with 773 PPI and it offers a frame rate of 60-120 Hz. The device creates a virtual grid, which is monitored by inside out tracking, that sets the boundaries in which the user can move.  DSI-24 is a wireless EEG headset with dry electrodes that is produced by Wearable Sensing, San Diego, CA, USA. …… The ECG electrodes were part of the DSI-24 bundle and are connected directly to the headset. This way, there is no need for synchronization of the EEG and ECG signals.”

Comment 2: The author should provide the method of analysis.

Response: We thank the reviewer for the valuable comment. The full methodology of the experimental protocol, the preprocesssing stage and the statistical analysis is described in the Materials and Methods section. Nevertheless, the first paragraph of Materials and Methods have been extended, according to the authors suggestion for better clarification. The following text has been added in Page 3, Line 121: “Our methodology includes three basic stages. The Recording, the Preprocessing and the Analysis stage. The former is the experimental and signal acquisition process stage, which in detail explained in section 2.2. In the second stage the signals are properly handled to be analyzed later by implementing techniques such as artifact removal, Heartbeat per Minute extraction etc. The details of this stage are laid out in section 2.3. The latter stage incorporates signal analysis techniques and statistical methodologies that investigate our scope/hypothesis and lead to our results and conclusions. Those can be found in sections 2.4 and 2.5”

Comment 3: In the Table 1, it is better that the authors label the significant result by using star sign instead of blackbody.

Response: We did as the reviewer instructed. In the reviewed version of the manuscript, statistical significance is now indicated with a * symbol, in all tables. Also, Table 1-3 captions have been modified and now explain the meaning of *.

Comment 4: In Figure 5, what does the plus sign mean?

Response: The + sign in these boxplots created in python indicate outlier values. There can be more than one in every boxplot (see OCC stress in the first plot). The caption of the Figure 5 has been modified according to the reviewer’s suggestion for better clarification.

Comment 5: In Figure 7, the authors should provide better quality figures.

Response: The figure has been enlarged for better readability, according to the reviewer’s suggestion.

Comment 6: The degree of stress should be provided.

Response: We thank the reviewer for this valuable comment. In the current study we use Heart Rate increase as a validator of stress. Thus, we may say that BPM increase is a measure of the degree of stress. However, stress is not quantifiable, nor any biomarker should be used directly as a meter of stress. We do not propose a scale of stress, but instead we create 2 different stress groups based on the BPM increase. Nevertheless, the Perceived Stress Scale is an established subjective measure of stress quantification. We obtained the responses of a Perceived Stress Scale questionnaire for every participant. However, due to the size of the participant pool at this stage, we decided not to use the results, but wait until more subjects have participated in the experiment. We will make use of these results in future publications of these experiment, when the participant pool will be significantly larger.

Round 2

Reviewer 1 Report

Thank you for addressing my concerns.

Only one concern is remained in the revised manuscript and your response.

For the response to my Comment 1, I think the authors' claim is inconsistent because the neural activity, namely EEG response also includes large inter-/intra-individual variances and they are often problematic as same as subjective ratings. The issue is not the PSS questionnaire, but all the variables you measured.

If you think the sample size was not enough for the PSS questionnaire, how did you determine the sample size of the experiment including the EEG/ECG measurements?  How much detectable difference was expected? Please clarify the sample size determination for the whole experiment.

In any case, the result of the PSS questionnaire should be included in the current paper for the clarity and reproducibility of the experimental process even if it did not follow your hypothesis. The distribution of PSS scores can be analyzed and whether the small sample size was a problem can also be discussed from that distribution. That is not inconclusive but informative for the readers. 

Author Response

Reviewer 1

Comment: For the response to my Comment 1, I think the authors' claim is inconsistent because the neural activity, namely EEG response also includes large inter-/intra-individual variances and they are often problematic as same as subjective ratings. The issue is not the PSS questionnaire, but all the variables you measured.

If you think the sample size was not enough for the PSS questionnaire, how did you determine the sample size of the experiment including the EEG/ECG measurements?  How much detectable difference was expected? Please clarify the sample size determination for the whole experiment.

In any case, the result of the PSS questionnaire should be included in the current paper for the clarity and reproducibility of the experimental process even if it did not follow your hypothesis. The distribution of PSS scores can be analyzed and whether the small sample size was a problem can also be discussed from that distribution. That is not inconclusive but informative for the readers. 

Response: We feel very thankful for your time and your determination to supporting our work. We and the reviewer initially had different opinions about the incorporation of the PSS scores. Indeed, we believe that a larger study would make better use of a subjective measure, since the distribution of a questionnaire’s scores may be alterated due to the small dataset. Regarding the reviewers question about clarifying the sample size determination, I would arbitrarily answer that, ideally more than double of the current participants would be required. We were not able to acquire such a participant pool due to equipment and university restrictions, and this is the reason for leaving the PSS aside.

However we found your objections correct, and we agree that any of our concerns should be expressed in the Discussion “limitations” section. So, we analyzed the PSS scores and performed some correlation analysis, finding some unexpected results.

Initially, we performed a Kolmogorov-Smirnov test that did not validated the normal distribution of the PSS results. Then, we performed Independent samples T-test and Mann-Whitney U-test to check if the PSS scores of the 2 groups were significantly different, but they were not. Finally, we performed Spearman correlation analysis to check for correlation between the values of the PSS with the values of BPM and brain region band powers. The results of the correlation analysis were quite interesting. High correlation between Beta and Gamma increase (from stressed state to calm state) and PSS score was observed. In other words, the participants who seemed to be stressful according to the PSS test, indeed shown increased brain activity at the stressful scenario. By searching the bibliography, we found that there is more research that presented similar correlation between EEG Beta band and PSS [1].

So, we did the following changes in the manuscript.

  • Figure 1 Flowchart has been revised
  • Materials and Methods has been revised. A Perceived stress scale section was added (section 2.2.1) , as well as a Statistical Analysis section (2.5)
  • The Results section was revised. Sub-sections Group Analysis (3.1) and Correlation Analysis (3.2) were created for better understanding of the reader. The results of the Mann-Witney U-test were added in Table 3 and described in section 3.1 The results of the correlation analysis of the PSS were added in a new table (Table 5) and analysed in section 3.2
  • A new paragraph was added in the discussion highlighting these results but making the reader aware of its possible limitations. We quote this paragraph.

“The correlation analysis between the PSS scores and the EEG biomarkers revealed some interesting results. There seemed to be a strong connection between the subjective self-reported stress quantification of the participants and the increase in the high frequency brain activity in all brain regions. Meaning that the participants that perceived themselves as stressful individuals had indeed increased brain activity during the stressful part of the experiment. Such results could further support the validity of using EEG biomarkers as stress detectors, considering that there is more evidence in previous research that PSS score and EEG band power under stress can be correlated [1], and should be taken into consideration in future research. However, these findings cannot stand credible on their own. Firstly, because we did not observe significant difference between the PSS score of the two groups as well as there was no correlation between the BPM and the PSS score (BPM being the established stress validator in this experiment). Secondly, because the size of the participant pool of this experiment, may have impacted the validity of the distribution of the self-ratings (the same experiment on a different group of people may or may not produce the same results). “

  • The results of the PSS scale along with the Gender, Age and Vr Experience information of the participants were added in Appendix.

References

  1. Hamid, N.H.A.; Sulaiman, N.; Murat, Z.H.; Taib, M.N. Brainwaves Stress Pattern Based on Perceived Stress Scale Test. In Proceedings of the 2015 IEEE 6th Control and System Graduate Research Colloquium (ICSGRC); IEEE, August 2015; pp. 135–140.

Reviewer 2 Report

The authors have addressed my most comments. However, there are still some issues need to be improved.

 1. In Figure 7, the information of axis-x and axis-y were missing.

2. Again, please carefully check whole manuscript. There are still some typing errors.

e.g. ‘Those can be found in sections 2.4 and 2.5’, the full stop was missing.

Author Response

Reviewer 2

Comment: The authors have addressed my most comments. However, there are still some issues need to be improved.

  1. In Figure 7, the information of axis-x and axis-y were missing.

Response: We thank the reviewer for the feedback. We have addressed your concern and added the Axis information in the label of the Figure 7. Specifically, we divided the Figure 7 into 3 parts (a,b,c instead of previously divided as a,b) and added Axis information separately for each of them.

Comment:  2.  Again, please carefully check whole manuscript. There are still some typing errors.  e.g. ‘Those can be found in sections 2.4 and 2.5’, the full stop was missing.

Response: We thoroughly checked the document for typing errors/mistakes and corrected them.

Moreover, we performed some major revisions after incorporating the PSS questionnaire results as suggested by reviewer 1. Detailed information can be found in the response of reviewer 1.